# Transforming Liver Cancer Therapy: Integrating Molecular Profiling with Precision and Transplant-Based Care

**DOI:** 10.3390/cancers17223671

**Published:** 2025-11-16

**Authors:** Seoung Hoon Kim

**Affiliations:** Organ Transplantation Center, National Cancer Center, 323 Ilsan-Ro, Ilsandong-Gu, Goyang-Si 10408, Gyeonggi-Do, Republic of Korea; kshlj@hanmail.net or kshlj@ncc.re.kr; Tel.: +82-31-9201647; Fax: +82-31-9201138

**Keywords:** liver transplantation, hepatocellular carcinoma, molecular profiling, precision medicine, immunotherapy, tumour microenvironment

## Abstract

Molecular and immunological profiling are transforming how hepatocellular carcinoma (HCC) is managed. Integrating biomarkers such as AFP, DCP, radiomics, and circulating tumour DNA refines transplant eligibility and post-transplant surveillance. Advances in understanding the tumour microenvironment and immune modulation guide the rational use of immunotherapy and mTOR-based immunosuppression. Liquid biopsy and multi-omics approaches enable earlier detection of molecular residual disease and tumour recurrence after curative treatment. Together, these developments support a precision-oriented, transplant-aware framework that bridges molecular discovery with clinical decision-making in everyday HCC care.

## 1. Introduction

Hepatocellular carcinoma (HCC) remains a high-burden malignancy with marked geographic and aetiological heterogeneity [1,2]. Curative-intent options—resection, ablation, and liver transplantation (LT)—are feasible only for a subset at presentation, and recurrence after locoregional or surgical therapy is common [3]. Although outcomes have improved with modern systemic therapy over the past decade, the prognosis for patients diagnosed at intermediate and advanced stages remains limited [4,5,6,7], reflecting both tumour biology and the constraints imposed by underlying liver dysfunction.

Systemic therapy has undergone a rapid shift from tyrosine-kinase inhibitor monotherapy to immune-based combinations [7]. First-line regimens that pair anti-angiogenic therapy with PD-(L)1 blockade, as well as dual immune–checkpoint strategies, now define the therapeutic landscape for many patients [4,5,6,7]. Yet real-world benefit varies widely due to portal-hypertension-related bleeding risk, performance status, and—crucially—biological heterogeneity [3,8]. Durable control hinges on aligning treatment with the molecular wiring of the tumour and the composition of the tumour microenvironment (TME), both of which shape response and resistance [9,10,11].

In parallel, transplantation practice is evolving. Broader adoption of downstaging protocols, refinement of allocation policies, and incorporation of serum markers have expanded access for selected patients beyond historical morphologic criteria [3,12]. Emerging tools—radiomics for microvascular invasion risk, 18F-fluorodeoxyglucose positron emission tomography (FDG-PET) avidity, multi-omics signatures, and liquid biopsy for molecular residual disease (MRD)—are beginning to inform selection before transplantation and surveillance after surgery [13,14,15,16,17]. These data also intersect with post-transplant management, including risk-adapted immunosuppression and cautious use of immunotherapy around the time of transplantation, where rejection risk must be balanced against potential oncological benefit [18,19,20,21,22].

This review adopts a translational perspective on these converging threads. I first summarise contemporary molecular and immune classifications of HCC and their clinical correlates, then examine how the TME shapes treatment response and resistance. I also evaluate liquid-biopsy and multi-omics approaches for surveillance and MRD detection and finally propose how these biomarkers can be integrated into pre- and post-transplant decision-making, including selection, downstaging, risk modelling, and immunosuppression strategies. My aim is to bridge basic science and clinical practice by providing a practical, precision-oriented, and transplant-aware framework for HCC management.

The overall organisation of this review is illustrated in Figure 1, which provides a schematic overview of its five major domains: molecular taxonomy, TME, systemic therapy, liquid-biopsy-based MRD monitoring, and transplant-based care.

Each domain represents a layer of the precision-and-transplant framework, showing how molecular and immune insights translate into patient selection, treatment optimisation, and post-transplant surveillance across the continuum of HCC management.

## 2. Molecular Taxonomy of HCC and Clinical Correlates

### 2.1. Canonical Gene-Expression Taxonomies (Hoshida, Boyault, Chiang)

Early transcriptome work resolved HCC into reproducible subclasses with distinct signalling and clinicopathologic features. Hoshida S1–S3 emerged from a cross-cohort meta-analysis (*n* = 603). S1 shows aberrant WNT/TGF-β activation and greater early recurrence, S2 is a proliferation phenotype [MYC/protein kinase B (AKT) targets] with larger tumours and higher serum AFP, and S3 is a hepatocyte-differentiated phenotype with lower AFP and better differentiation. These labels correlated with size, grade, AFP and early recurrence risk across datasets [23].

In parallel, Boyault G1–G6 linked unsupervised clusters to driver alterations and aetiology. G1/G2 associate with HBV and AKT/PI3K activity (G2 enriched for TP53/PIK3CA mutations). G3 is TP53 mutated with cell-cycle activation, G4 includes TCF1-mutated tumours, and G5/G6 are CTNNB1/β-catenin classes with canonical WNT activation (G6 with E-cadherin loss and satellite nodules) [24].

A third scheme from Chiang et al. defined five classes—CTNNB1, proliferation, IFN-related, a chromosome-7 polysomy class, and an unannotated group—again aligning expression with copy-number and mutation data (e.g., CTNNB1 mutations enriched in the CTNNB1 class; IGF/AKT-mTOR signalling in proliferation class) [25].

Taken together, these frameworks converge on clinically meaningful axes such as proliferation vs. differentiation, WNT/β-catenin activity, progenitor/stemness traits, and have been externally summarised and applied to independent cohorts (Table 1) [26].

### 2.2. A Clinically Translatable, Integrative View

Subsequent synthesis places HCC along two broad, practice-oriented archetypes.

The non-proliferation class comprises well-differentiated tumours, often harbouring CTNNB1 (β-catenin) mutations and displaying metabolic or hepatocytic signatures with low AFP levels. In contrast, the proliferation class features dedifferentiated biology with frequent TERT-promoter activation, TP53 mutation, FGF19/FGFR4-driven growth-factor signalling, chromosomal instability, and higher AFP levels. This integrative dichotomy distils earlier gene-expression taxonomies and remains widely used in translational and clinical review frameworks [27,28].

Pan-omic studies reinforced the gene-level foundations: TCGA (363 cases) catalogued recurrent mutations (notably TERT, CTNNB1, TP53) and identified integrative subtypes across platforms; focal FGF19 amplification on 11q13.3 mechanistically links to FGFR4 signalling as a druggable driver in a molecular subset [29,30,31].

### 2.3. Proteogenomics: Transcript–Protein Discordance and Pathway-Level Rewiring

Proteogenomic atlases extend this framework by quantifying how signalling is rewired beyond DNA/RNA.

An ICPC/Cell study of HBV-related HCC (*n* = 159) resolved proteome subclasses (metabolism, proliferation, microenvironment) and highlighted discordance between mRNA and protein, with metabolism and kinase signalling (e.g., AKT–mechanistic target of rapamycin [mTOR]) reprogrammed at the protein/phosphoprotein level [32].

A broader Nature Communications analysis across aetiologies and stages showed only moderate transcript–proteome correlation (Spearman’s ρ = 0.33), and mapped alterations in the WNT–β–catenin, AKT/mTOR, Notch and cell-cycle pathways at genomic, proteomic and phosphoproteomic layers; it also tied CTNNB1 and TP53 mutations to distinct phospho-signalling outputs [33].

Mechanistic and review work around HBV-HCC further contextualises these findings and points to proteogenomic candidates (e.g., cholesterol/FA metabolism, SOAT1) with therapeutic implications [34].

 Concluding Remarks 

Legacy molecular taxonomies such as the Hoshida (S1–S3), Boyault (G1–G6), and Chiang classifications converge into a clinically applicable dichotomy of proliferation versus non-proliferation phenotypes. Within this framework, specific oncogenic drivers such as CTNNB1 and FGF19/FGFR4 refine biological risk assessment and generate targeted therapeutic hypotheses. Proteomic and phosphoproteomic analyses further demonstrate that expression-based signatures do not always correspond to protein activity, highlighting the importance of multi-omics profiling whenever feasible.

## 3. Tumour Microenvironment (TME) and Immune Modulation

### 3.1. Single-Cell Cartography and Immunosuppressive Niches in HCC

Single-cell atlases consistently show that HCC harbours myeloid-dominated, tolerogenic ecosystems. Across cohorts, TREM2^+^ tumour-associated macrophages (TAMs) accumulate within tumours, correlate with worse outcomes, and exhibit ligand–receptor programmes that recruit Tregs and dampen cytotoxic T-cell function [11,35,36,37]. Functionally, in patient-derived and murine HCC models, TREM2 loss restored CD8^+^ T-cell infiltration and potentiated anti-PD-L1 efficacy, nominating TREM2^+^ TAMs as actionable suppressors (with particular enrichment after TACE) [36]. LAMP3^+^ dendritic cells (mregDCs)—CCR7^+^, migratory DCs that can co-express regulatory molecules—are prominent in HCC single-cell maps. Although context-dependent, they often align with exhausted T-cell states and immunotherapy-relevant antigen-presentation circuits [9,11,38]. These features are concordant with MDPI syntheses of the HCC TME, which emphasise the liver’s basal tolerance and myeloid skew as barriers to immune control [38]. Therapeutically, these myeloid and dendritic-cell subsets are now being explored as targets for immune modulation. TREM2 blockade or myeloid reprogramming strategies aim to restore antitumour immunity, while modulation of LAMP3^+^ dendritic-cell circuits may enhance responsiveness to immune checkpoint inhibitors (ICIs).

### 3.2. WNT/CTNNB1 Activation, Immune Exclusion, and ICI Resistance

Mechanistically, tumour-intrinsic WNT/β-catenin activation blocks recruitment of BATF3-lineage dendritic cells, eliminating the chemokine cues needed for priming and intra-tumour T-cell entry (a “non-T-cell-inflamed” state) across cancers [39,40]. In HCC, β-catenin activation drives anti-PD-1 resistance in engineered mouse models and has been linked to non-response in early clinical series, establishing CTNNB1-mutant HCC as a biomarker-defined immune-excluded class [9,10].

Translational reviews and MDPI updates converge on this framework and argue for β-catenin–aware trial design (e.g., avoiding ICI monotherapy in CTNNB1-mutant disease; testing priming strategies that restore DC influx) [39].

### 3.3. Emerging Targets and Combination Strategies

#### 3.3.1. Adenosine–CD73–A2A/A2B Axis

Adenosine signalling is a central, druggable suppressive circuit in HCC. A2A-receptor blockade augments anti-PD-(L)1 activity in HCC murine models, improving T-cell function and tumour control [41]. Clinically, CD73 (NT5E) is over-expressed in a sizeable subset of HCC and associates with metastasis and poor prognosis, and early-phase anti-CD73 antibodies (e.g., oleclumab) have shown immune activation and tolerable safety in solid tumours (with combinatorial signals in non-HCC settings), motivating HCC-specific trials [42,43,44]. Importantly, aetiology matters: in NASH-associated HCC models, A2A signalling may exert tumour-suppressive effects, cautioning against one-size-fits-all inhibition and underscoring the need for biomarker-guided selection [45]. Comprehensive reviews outline how A2A/A2B and CD73 cooperate to enforce myeloid and endothelial suppression in liver tumours and propose rational PD-(L)1±VEGF combinations [46].

#### 3.3.2. LAG-3 and TIGIT Checkpoints

In HCC, TIGIT combinations have yielded mixed early clinical signals. The randomised MORPHEUS-Liver phase Ib/II study suggested higher ORR when tiragolumab (anti-TIGIT) was added to atezolizumab+bevacizumab versus the doublet alone, prompting further evaluation [47]. In LIVERTI (phase II), domvanalimab + zimberelimab achieved an ORR of 17.2% in PD-(L)1-refractory HCC, with manageable toxicity and exploratory ctDNA pharmacodynamics, but did not meet its pre-specified threshold—supporting activity yet highlighting the need for better selection [48]. For LAG-3, clinical experience in HCC is nascent (e.g., nivolumab + relatlimab trials), while preclinical work combining anti-LAG-3 + anti-PD-1 + STING agonism shows synergistic suppression and enables LAG-3–targeted PET as a pharmacodynamic read-out [49]. Together, the data argue for context-specific deployment of next-generation checkpoints and for embedding correlative myeloid/DC biology in trials.

#### 3.3.3. Novel TAM Subsets (Early Signals)

Beyond TREM2^+^ macrophages, a CD19^+^ TAM subpopulation has recently been described in a 2025 preprint (arXiv:2503.17738). These cells were reported to express PD-L1 and CD73 and to be targetable by anti-CD19 CAR-T in preclinical models. While intriguing, these findings remain preliminary and non-peer-reviewed and thus require cautious interpretation and independent validation before any clinical translation can be considered.

Concluding Remarks

In practice, myeloid-inflamed phenotypes (TREM2^+^/SPP1^+^ TAM programmes, characterised by paucity of cDC1) track with primary resistance to PD-(L)1 and early relapse, whereas DC-rich ecosystems, tertiary lymphoid structure formation, and intact cross-presentation favour responsiveness.

## 4. Precision Systemic Therapy Landscape (2022–2025)

First-line systemic therapy for unresectable or advanced hepatocellular carcinoma currently relies on three immunotherapy-based combinations that have replaced sorafenib monotherapy as the historical standard. These regimens—atezolizumab plus bevacizumab (IMbrave150), tremelimumab plus durvalumab (HIMALAYA/STRIDE), and nivolumab plus ipilimumab (CheckMate 9DW)—are summarised below.

 First-Line Therapy—Where We Are Now. 

The IMbrave150 regimen (atezolizumab + bevacizumab) established a new standard by improving overall survival (OS) and progression-free survival (PFS) over sorafenib; with longer follow-up, median OS was 19.2 vs. 13.4 months (HR 0.66) and PFS 6.9 vs. 4.3 months (HR 0.65) [4,7]. The combination received FDA approval in May 2020 for systemic-therapy-naïve, unresectable/metastatic HCC. Because bevacizumab increases bleeding risk, selection requires assessment and management of portal-hypertension complications; contemporary guidance recommends endoscopic screening/treatment of varices before therapy, and the IMbrave150 protocol mandated upper endoscopy within 6 months of initiation [50,51].

The STRIDE regimen (single priming dose tremelimumab + durvalumab, then durvalumab maintenance) improved OS versus sorafenib (HR ≈ 0.78) and delivered durable long-term survival (e.g., 4–5 year OS plateaus in updated analyses), leading to FDA approval in October 2022 [5,52]. STRIDE is a VEGF-inhibitor-free and is often preferred when bleeding risk or other contraindications to anti-angiogenesis exist, with the trade-off of higher rates of immune-related adverse events due to cytotoxic T-lymphocyte–associated protein 4 (CTLA-4) priming [5,52].

In 2025, CheckMate-9DW reported that nivolumab + ipilimumab improved OS versus investigator’s choice lenvatinib or sorafenib (median 23.7 vs. 20.6 months; HR 0.79). Kaplan–Meier curves crossed early (more deaths in the first ~6 months in the nivolumab + ipilimumab arm) but separated thereafter, supporting a survival tail in selected patients. The regimen subsequently received FDA approval for first-line unresectable HCC [53]. These three options—atezolizumab + bevacizumab, STRIDE, and nivolumab + ipilimumab—now anchor first-line decision-making; choice is guided by portal-hypertension/bleeding risk, autoimmune comorbidity, and clinician–patient preference regarding toxicity profiles [5,50,53].

Second-Line Therapy—What Still Works.

After progression on/unsuitability for first-line immune-oncology (IO)–based combinations, multikinase inhibitors and VEGFR2 blockade remain evidence-based standards. In RESORCE, regorafenib (sorafenib-tolerant progressors) improved OS to 10.6 vs. 7.8 months (HR 0.63) and is FDA-approved in this setting [54]. In CELESTIAL, cabozantinib improved OS to 10.2 vs. 8.0 months (HR 0.76) and is FDA-approved after sorafenib [55]. For patients with AFP ≥ 400 ng/mL, ramucirumab (REACH-2) improved OS (HR ~ 0.71) and is FDA-approved; it is the first biomarker-selected therapy validated in phase III HCC [56]. Practical sequencing today commonly uses one of these agents after first-line IO-based therapy, taking liver functional reserve and prior tolerance into account.

Comparative outcomes from pivotal systemic-therapy trials are summarised in Table 2, highlighting overall- and progression-free-survival gains, hazard ratios, and key toxicity profiles across first- and second-line regimens.

Moving IO into the Intermediate Stage (TACE-combination trials).

Two phase III trials tested whether adding systemic IO (±VEGF/VEGFR blockade) to TACE improves outcomes. EMERALD-1 (durvalumab + bevacizumab + TACE vs. TACE + placebo) met its primary endpoint: PFS improved (median 15.0 vs. 8.2 months; HR 0.77); OS remains immature. Safety was manageable and consistent with components [57]. LEAP-012 (lenvatinib + pembrolizumab + TACE vs. TACE + placebo) also met its PFS primary endpoint (median 14.6 vs. 10.0 months; HR 0.66), with a numerical OS advantage not yet mature; grade ≥ 3 treatment-related AEs were higher with the triplet (71% vs. 32%) and included rare treatment-related deaths, underscoring the need for careful patient selection and multidisciplinary monitoring [58]. Together, these data support PFS benefit from IO- or IO/TKI-augmented TACE, while definitive OS and long-term hepatic safety signals will determine guideline uptake.

Biomarker-Driven Therapy—the Reality Check (2022–2025).

Tissue-agnostic indications apply to a small minority of HCC: pembrolizumab is approved for MSI-H/dMMR tumours (2017) and for TMB-high (≥10 mut/Mb) tumours (2020), and TRK inhibitors (larotrectinib, entrectinib) are approved for NTRK fusion–positive tumours. However, in large genomic series of advanced HCC, MSI-H and TMB-H are rare (≈0.2% and ≈0.8%, respectively), and NTRK fusions are exceedingly uncommon (<1% across most common tumours; pan-cancer prevalence ≈ 0.3%)—so biomarker-matched, tissue-agnostic therapies remain opportunistic rather than routine in HCC [59,60].

FGF19/FGFR4 Axis—Lessons from Target Development.

The FGF19–FGFR4 pathway defines a biologically coherent HCC subset. Early-phase trials of selective FGFR4 inhibitors (e.g., fisogatinib/BLU-554; roblitinib/FGF401) showed activity in FGF19-positive or FGFR4/KLB-positive tumours, validating pathway dependence but not yet yielding a registrational success [61,62]. Mechanistically, on-target resistance (e.g., FGFR4 gatekeeper/hinge-region mutations) and adaptive bypass via EGFR-MAPK reactivation have been described, suggesting rational combinations (e.g., EGFR plus FGFR4 blockade) and refined biomarker selection as future directions [63,64,65].

Concluding Remarks

First-line selection should be individualised. Atezolizumab–bevacizumab offers the most mature survival benefit but requires pre-treatment variceal screening and management and is unsuitable where bleeding risk is high. STRIDE (tremelimumab–durvalumab) is VEGF-inhibitor-free and is preferred when anti-angiogenesis is contraindicated, accepting higher immune-related toxicity with CTLA-4 priming. Nivolumab–ipilimumab provides a survival tail with an early hazard trade-off and suits patients without pressing bleeding risk or uncontrolled autoimmunity. After progression on IO-based therapy, regorafenib and cabozantinib remain standards, and ramucirumab benefits patients with AFP ≥ 400 ng/mL. In the intermediate stage, IO/TACE “triplets” improve PFS, but OS remains immature and grade ≥ 3 toxicity is higher, warranting careful selection and multidisciplinary monitoring. Tissue-agnostic options (MSI-H, TMB-high, NTRK fusions) apply to a small minority of HCC. FGFR4-directed agents show activity in FGF19-positive tumours, yet resistance mechanisms argue for refined biomarkers and rational combinations.

## 5. Adjuvant and Neoadjuvant Therapy

### 5.1. Where the Field Stands in 2025: Adjuvant Immunotherapy After Resection/Ablation

The phase III IMbrave050 trial initially reported an improvement in recurrence-free survival (RFS) with atezolizumab + bevacizumab versus active surveillance after curative-intent resection or ablation in high-risk HCC (HR 0.72 at the first interim analysis) [8,66]. However, the updated analysis (median follow-up 35.1 months) showed the RFS benefit was not sustained (stratified HR 0.90; 95% CI 0.72–1.12), and overall survival (OS) remained immature, with >80% alive in both arms at 2 years. On the basis of these data, the AASLD 2025 Critical Update advises against the use of adjuvant systemic therapy (including atezolizumab + bevacizumab) outside a clinical trial and similarly discourages neoadjuvant systemic therapy for patients undergoing resection/ablation [8].

History underscores this caution. In STORM, adjuvant sorafenib did not improve RFS versus placebo after resection or ablation (HR 0.94; 95% CI 0.78–1.13) and added toxicity, reinforcing surveillance as the standard after curative therapy [67].

As of September 2025, there are no FDA-approved adjuvant (or neoadjuvant) systemic therapies for HCC. Surveillance remains the standard after resection/ablation, and (neo)adjuvant strategies should be pursued only in trials [8].

### 5.2. Signals from Neoadjuvant/Perioperative Immunotherapy (Hypothesis-Generating)

Across prospective studies and pooled analyses, neoadjuvant immune checkpoint blockade elicits meaningful pathological regression in a subset of resectable HCC.

A cross-trial, patient-level analysis (NeoHCC consortium; *n* = 111) reported major pathological response (MPR) in 32% and pathologic complete response (pCR) in 18%, with deeper regression associated with improved relapse-free survival [68].

In a randomised perioperative phase II study, nivolumab with or without ipilimumab was feasible, achieving MPR and enabling timely resection, though sample sizes were modest and long-term outcomes remain exploratory [69].

Similarly, a phase Ib trial of neoadjuvant cabozantinib plus nivolumab converted a proportion of borderline-resectable cases to R0 resection with immunologic remodelling of the TME [70].

Collectively, these studies justify continued investigation but do not yet support routine use; the AASLD update explicitly recommends against neoadjuvant systemic therapy outside trials [8].

### 5.3. Ongoing and Recently Reported (Neo)Adjuvant Trials (Overview)

Typical initiation of adjuvant therapy in trials: 4–8 weeks post-surgery/ablation, contingent on recovery and liver function [71]. As summarised in Table 3, several phase Ib–III studies are evaluating perioperative immunotherapy and targeted strategies in resectable or locally advanced HCC.

### 5.4. Practical Interpretation for Clinicians

Post-resection or ablation management should focus on surveillance, as routine adjuvant or neoadjuvant systemic therapy is not recommended outside clinical trials. Participation in (neo)adjuvant trials is encouraged for patients at high risk of recurrence, typically 4–8 weeks after surgery or ablation once hepatic function has recovered. Given the non-proportional hazards observed in IMbrave050, future trials should include robust interim rules and incorporate overall survival (OS), patient-reported outcomes, and biomarker-guided selection to prevent over-interpretation of early RFS signals.

Concluding Remarks

Despite a strong biological rationale, no adjuvant or neoadjuvant systemic therapy has yet demonstrated a sustained benefit after curative resection or ablation in HCC. Surveillance; therefore, remains the standard of care following curative treatment, and perioperative immunotherapy should currently be pursued only within clinical trials. Ongoing phase III studies will clarify whether biomarker-guided or combination approaches can redefine postoperative management in the coming years.

## 6. Liquid Biopsy and Multi-Omics for Surveillance and MRD

### 6.1. Rationale and Clinical Use-Cases

Despite curative-intent resection, ablation, or LT, MRD frequently precedes radiographic recurrence in HCC. Circulating tumour DNA (ctDNA) enables serial, minimally invasive detection of MRD and dynamic risk stratification that can outperform AFP alone and, in some series, predate imaging by months [72]. Early HCC data now span post-resection and post-LT settings, with growing—though still heterogeneous—evidence of prognostic and lead-time value [72,73,74].

Post-resection surveillance studies show that tumour-informed or plasma-only assays can detect MRD and predict relapse earlier than conventional markers [73,74].

Post-transplant surveillance data also suggest that ctDNA positivity correlates with subsequent recurrence, with important analytical caveats unique to LT, such as interference from donor-derived cell-free DNA (cfDNA) [16,75].

### 6.2. Assay Modalities and Readouts

Variant-based assays

Two paradigms are used in HCC MRD studies: tumour-informed (patient-specific variants tracked in plasma) and tumour-naïve (fixed panels). In a plasma-only, tumour-naïve approach, ctDNA positivity after curative resection predicted early relapse in HCC (prospective cohorts; analytical pipeline tailored to HCC mutational spectra) [73]. A postoperative ctDNA-TMB readout from a 47-patient series independently predicted recurrence and outperformed AFP (AUC 0.752) [76]. In a 125-patient real-world HCC analysis using a tumour-informed assay, serial ctDNA monitoring detected recurrence earlier than AFP, with a median lead time ~7.9 months in subcohorts (post-resection and post-LT) and clear prognostic separation by ctDNA status [74].

Epigenetic assays (5mC/5hmC)

Because tumour fraction is often low in early or minimal disease, methylome and hydroxymethylome (5hmC) signals can enhance sensitivity. Targeted enzymatic methyl-seq (EM-seq) panels have achieved AUC ~0.96 for HCC detection across stages (sensitivity ~90%, specificity ~97%), illustrating technical feasibility for low-input cfDNA and the potential for MRD applications as panels mature [77]. A newer library strategy, NEEM-seq (No End-repair Enzymatic Methyl-seq), avoids end-repair artefacts in cfDNA and, combined with a trained neural network (DeepTrace), delivered ~93.6% sensitivity and ~98.5% specificity for HCC detection in a validation cohort (BCLC 0/A sensitivity ~89.6%)—important proof-of-concept for ultra-low-fraction detection [78]. EM-seq’s underlying chemistry (enzymatic conversion preserving 5mC/5hmC) offers advantages over bisulfite methods for fragmented cfDNA [79]. Beyond 5mC, 5hmC signatures (e.g., 32-gene panels) have repeatedly shown value for early HCC detection, with potential to transition into MRD monitoring frameworks as longitudinal datasets accumulate [80,81].

Fragmentomics/multi-omics.

Fragmentome features (size profiles, end motifs, nucleosome footprints) can complement methylation to boost sensitivity at very low tumour fractions; liver cancer has been a leading testbed for such multi-omic cfDNA classifiers [82].

### 6.3. Evidence Summary by Setting

In post-resection HCC, tumour-naïve plasma-only NGS approaches can detect MRD and stratify relapse risk after R0 resection [73].

Post-operative ctDNA tumour mutational burden (TMB) positivity is associated with shorter recurrence-free survival and a higher recurrence risk compared with AFP-based monitoring [76].

Tumour-informed serial monitoring in real-world cohorts demonstrates earlier molecular recurrence than AFP or imaging, with clinically actionable lead times of approximately 7.9 months [74].

In pre- and post-transplant HCC, programmatic series integrating ctDNA testing across liver malignancies—including HCC—show higher post-transplant recurrence among ctDNA-positive patients and document ctDNA clearance after LT in some cases, consistent with curative systemic debulking by total hepatectomy [16].

A dedicated serial post-LT HCC study found that ctDNA positivity effectively identified MRD and predicted recurrence during surveillance [75].

### 6.4. Practical Interpretation and Integration

Data from HCC-focused and pan-cancer MRD studies suggest that q6–12-week sampling during the first 12–18 months post-curative therapy can detect molecular recurrence earlier than standard imaging intervals, as illustrated in Figure 2 [74,83]. In tumour-informed HCC cohorts, the median lead-time vs. AFP was ~8 months, supporting the use of treatment-triggering or imaging-escalation algorithms when ctDNA is positive [74]. Pan-cancer plasma-only MRD reports show an average of ~7–9 months lead time to radiographic relapse, a useful design anchor when adapting protocols locally [84].

In post-resection HCC, persistent or rising ctDNA (or ctDNA-TMB detection) indicates high relapse risk, warranting intensified imaging, trial referral, or adjuvant-style strategies where available.

In the post-transplant setting, ctDNA results should be interpreted in the context of donor-derived cfDNA (dd-cfDNA) assays and graft events. A true tumour-derived signal should be corroborated by orthogonal assay or imaging, given post-LT biology and potential analytical interference [85,86,87].

### 6.5. Limitations and Pitfalls

Low shedding and aetiology-related noise can limit assay sensitivity. Early or microscopic disease in HCC may release minimal ctDNA, and necroinflammatory liver disease can further complicate background cfDNA and methylation signals [72].

Clonal hematopoiesis (CHIP) may generate false-positive variant calls due to mutations in genes such as DNMT3A, TET2, ASXL1, and TP53. The best practice is to perform matched WBC sequencing or apply robust bioinformatic CHIP filtering, as recommended by expert guidelines [88].

Methylation-based assays can also yield false positives. Non-malignant conditions, including viral hepatitis, may induce aberrant methylation patterns that mimic tumour-derived signals in plasma-only analyses; careful clinical correlation and confirmatory testing are required [89].

Transplant-specific interference represents another challenge. Donor-derived cfDNA increases with rejection, ischaemia–reperfusion injury, or graft complications and can confound assay interpretation. Teams should coordinate ctDNA and dd-cfDNA workflows, use tumour-specific (rather than donor-derived) markers, and interpret results alongside graft status [85,86,87].

Finally, standardisation and access remain barriers to broad implementation. Cut-offs, reporting units (mut/Mb, MTM/mL), preanalytical factors, and reimbursement vary across studies, underscoring the need for multicentre prospective trials to define actionable thresholds and MRD-triggered interventions in HCC [72].

Concluding Remarks

Variant-based ctDNA assays currently provide the most mature signal for MRD detection in HCC, with growing evidence of lead time and prognostic value after resection and signal-positive associations after LT.

Methylome and 5hmC assays, as well as fragmentomic approaches, offer the greatest near-term sensitivity gains for detecting ultra-low tumour fractions, while enzymatic sequencing methods such as EMseq and NEEMseq help mitigate technical artefacts.

In clinical practice, ctDNA can serve as a risk-enrichment and surveillance adjunct while ongoing trials clarify MRD-guided therapy strategies. In LT recipients, ctDNA assessment should be interpreted within transplant-aware algorithms that also incorporate donor-derived cfDNA monitoring.

## 7. Transplant-Based Care: Selection, Downstaging, and Allocation

### 7.1. BCLC 2022: Where Transplant Fits

The Barcelona Clinic Liver Cancer staging system (BCLC) 2022 update explicitly recognises BCLC-B heterogeneity, encourages treatment stage migration in selected patients, and situates transplantation as a consideration in intermediate disease when tumour biology and liver function predict benefit (e.g., after effective downstaging) [3]. This framework, coupled with centre-level expertise and donor availability, underpins a more flexible, biology-aware pathway to LT beyond historical morphology-only gates [3].

### 7.2. Policy and Allocation (OPTN/UNOS): Standardised Exception, Downstaging, and Li-Rads Harmonisation

In the United States, OPTN/UNOS Policy 9.5.I govern HCC MELD exception and downstaging.

The T2 definition, which determines eligibility for a standardised exception, includes either one Class 5 lesion 2–5 cm or two to three Class 5 lesions 1–3 cm, with serum AFP ≤ 1000 ng/mL [90].

Downstaging eligibility (UNOS DS) before LRT allows one Class 5 lesion 5–8 cm, two or three lesions (≥1 >3 cm, each ≤ 5 cm, sum ≤ 8 cm), or four to five lesions each < 3 cm with a total diameter ≤ 8 cm. After LRT, viable disease must meet T2 criteria on dynamic CT or MRI to qualify for a standardised exception [90].

When AFP > 1000 ng/mL, candidates may be treated, but an exception requires AFP < 500 ng/mL and sustained control below that level; otherwise, cases go to the NLRB and are generally not suitable for exception listing [12,90].

Since 13 July 2023, OPTN/UNOS guidance has been aligned with LI-RADS terminology, and contrast-enhanced ultrasound (CEUS) is now accepted as a diagnostic tool for Class 5 observations, with policy tables updated accordingly [90,91].

These changes standardise imaging language, codify UNOS-DS inclusion, and add AFP-based safeguards against aggressive biology, as summarised in Table 4 [90,91].

### 7.3. Indications and Outcomes: Deceased-Vs Living-Donor; Downstaging Performance

Deceased-donor LT (DDLT). In the first prospective, multiregional study using UNOS-DS criteria, successful downstaging to within Milan exceeded 80%, with 2-year post-LT survival ~95% and outcomes comparable to patients within Milan at listing [92]. Earlier cohorts also showed that successful downstaging to T2 yields low recurrence and excellent survival after LT [93]. 10-year longitudinal data confirm durable outcomes when downstaging is achieved (e.g., 5-year RFS ~87% in contemporary series) [94].

Living-donor LT (LDLT). LDLT programmes often apply expanded biology-integrated criteria, with favourable outcomes when surrogate markers (e.g., AFP/DCP) are controlled and response to LRT is demonstrated [95,96]. The Kyoto criteria—≤10 tumours, each ≤ 5 cm, and DCP (PIVKA-II) ≤ 400 mAU/mL—identify LDLT candidates with low recurrence comparable to Milan-in-criteria recipients [97,98].

Beyond “within criteria”, radiological response (modified Response Evaluation Criteria in Solid Tumours, mRECIST) before LT refines risk: incorporating mRECIST into Metroticket 2.0 improves prediction of HCC-related death and supports stricter selection when partial response (PR)/stable disease (SD) or progressive disease (PD) persists after locoregional therapy (LRT) [99,100].

### 7.4. Biomarker-Integrated Selection Models (Schematic Summary)

The French AFP model (AFP score) combines tumour number, largest diameter, and AFP level (0–9 points) to predict post-LT recurrence. Its performance has been validated and adopted in France to replace the Milan criteria in allocation practice [101,102].

Metroticket 2.0 is a competing-risk model that incorporates AFP, tumour size, and number to estimate 5-year survival and HCC-related death, with a publicly available calculator. Incorporating mRECIST response further improves prognostic discrimination after LRT [99,103].

The Kyoto criteria (LDLT) define eligibility as ≤ 10 nodules, each ≤ 5 cm, and DCP ≤ 400 mAU/mL. This expanded yet biology-aware set has demonstrated low recurrence rates in validation cohorts [97,98].

The up-to-Seven rule—the sum of the largest tumour diameter (cm) and the number of tumours being ≤ 7—is widely used as an expanded morphologic gate, achieving acceptable 5-year survival when tumour biology is favourable or well controlled [104,105].

Together, these models shift transplant candidacy assessment from morphology alone to an integrated framework encompassing morphology, biology, and treatment response, aligning with the BCLC emphasis on stage migration in carefully selected patients [3].

### 7.5. Imaging, Radiomics, and FDG-PET for Microvascular Invasion (MVI) and Recurrence Risk

Microvascular invasion (MVI) is a major determinant of post-transplant recurrence but can only be confirmed histologically. Consequently, accurate pre-operative prediction is crucial to optimise candidate selection, bridging strategies, and post-transplant surveillance.

Meta-analyses and systematic reviews demonstrate moderate-to-good accuracy of CT and MRI-based radiomics for pre-operative MVI prediction, with clinicoradiomic models consistently outperforming traditional clinico-radiologic approaches in many studies [13,14].

Across surgical and transplant cohorts, increased 18F-FDG uptake on PET correlates with MVI, poor tumour differentiation, and higher post-transplant recurrence, thereby identifying aggressive tumour biology beyond conventional morphology [106,107,108]. Contemporary research continues to refine quantitative PET parameters, such as SUVmax and tumour-to-liver ratios, which hold prognostic value before transplantation [108].

Concluding Remarks

Liver-transplant selection for HCC has shifted from morphology-based criteria toward an integrated framework that combines morphology, biology, and treatment response.

Modern allocation policies (such as UNOS-DS), dynamic biomarkers (AFP, DCP, AFP-L3), and advanced imaging modalities—including radiomics and FDG-PET—now enable risk-adapted selection beyond static size and number limits.

Incorporating validated models such as the AFP score, Metroticket 2.0, Kyoto criteria, and up-to-seven improves prognostic precision and aligns with the BCLC 2022 emphasis on biology-driven stage migration.

Continued harmonisation of policy and biology-anchored tools will further enhance equity, transparency, and post-transplant outcomes in HCC management.

## 8. Integrating Biomarkers into Transplant Decisions

Integrating serologic, imaging, and genomic biomarkers into transplant decision-making allows selection and surveillance to be refined according to tumour biology rather than morphology alone. Key biomarker-integrated tools and post-transplant risk models are summarised in Table 5.

### 8.1. Pre-LT “Risk Biology” to Steer Selection, Waiting, and Downstaging

Routine morphology (e.g., Milan/UNOS-DS) increasingly shares the stage with biology-anchored markers. Serum AFP dynamics and composite tumour markers refine candidacy: US allocation policy withholds an automatic exception when AFP > 1000 ng/mL unless it can be reduced and sustained < 500 ng/mL, reflecting aggressive biology and a higher risk of wait-list drop-out. Programmes should document treatment response before exception approval [12]. DCP (PIVKA-II) and AFP-L3 add orthogonal risk information—dual-positive profiles are consistently linked to early recurrence and poorer post-LT outcomes, supporting their use for intensifying bridging therapy and imaging while on the list [109]. In LDLT settings, the Kyoto criteria (≤10 tumours, each ≤5 cm, DCP ≤ 400 mAU/mL) exemplify biology-integrated selection with low post-LT recurrence across validations [98].

Non-invasive imaging phenotypes also stratify “risk biology.” Pre-LT 18F-FDG PET avidity correlates with microvascular invasion (MVI), poor differentiation, and higher post-LT recurrence. PET; therefore, it helps flag candidates who may need more stringent downstaging or extended observation despite morphologic eligibility [106]. Complementing PET, radiomics from CT/MRI offers pre-operative MVI prediction with moderate-to-good performance in meta-analyses, and clinicoradiomic models often outperform clinico-radiologic baselines—useful where biopsy is not feasible [13,14]. Transcriptional signatures are emerging; biopsy-based MVI gene panels (e.g., six-gene models) and LT-focused signatures show promise for forecasting post-LT recurrence and could be layered onto imaging and serology in research protocols. At present, they remain investigational for routine listing decisions [110,111].

The pre-LT selection process can be applied through a tiered framework. Candidates should first meet morphologic criteria consistent with established transplant guidelines. Favourable biology should then be confirmed—typically indicated by falling or low AFP levels, single-positive rather than dual-positive DCP/AFP-L3, PET non-avid or only mildly avid uptake, and radiomics not strongly predictive of microvascular invasion. A demonstrable bridging or downstaging response is required before an exception can be granted. When biological and morphologic markers are discordant—for example, cases that meet Milan criteria but are PET-avid with rising AFP—centres should consider extended observation, intensified locoregional therapy, or the use of LDLT criteria that explicitly integrate biological factors, such as the Kyoto model.

**Table 5 cancers-17-03671-t005:** Biomarker-integrated selection and post-LT risk tools.

Tool	Variables/Cut-offs	Output/Use	Notes	Refs.
AFP score (French model)	No. tumours; largest diameter; AFP (0–9 points)	Recurrence risk; allocation in France	Validated; improves over Milan	[100,101]
Metroticket 2.0	Size; number; AFP	5-year survival/HCC death (competing risks)	mRECIST response further improves discrimination	[98,102]
Kyoto (LDLT)	≤10 nodules, each ≤5 cm; DCP ≤ 400 mAU/mL	Expanded LDLT gate with low recurrence	Biology-aware expansion	[96,97]
Up-to-Seven	Largest tumour (cm) + number ≤ 7	Expanded morphologic gate	Outcomes depend on biology/response	[103,104]
RETREAT	AFP at LT + MVI + viable burden (diameter + number)	5-year recurrence tiers (<3% → >75%)	Externally and prospectively validated	[111,112,113]
mRETREAT	RETREAT + AFP-L3, DCP	Improved AUC/calibration	Early data; thresholds †: AFP-L3 ≥ 15%, DCP ≥ 7.5 ng/mL	[114,115]

† Thresholds from single-centre development/validation; calibrate locally.

### 8.2. Post-LT Risk Stratification and Surveillance

The Risk Estimation of Tumour Recurrence After Transplant (RETREAT) score—comprising AFP at LT, MVI, and the sum of the largest viable tumour diameter and number—is the most widely validated post-LT tool. It stratifies 5-year recurrence risk from less than 3% (score 0) to more than 75% (score ≥ 5) and guides the intensity of surveillance [112]. External validations in Europe and North America confirm discrimination, with particularly high negative predictive value in within-Milan cohorts [113]. In 2025, a prospective multicentre study further validated RETREAT’s accuracy and feasibility for programme-level surveillance pathways [114]. Building on this, the modified Risk Estimation of Tumour Recurrence After Transplant (mRETREAT) score augments RETREAT with AFP-L3 and DCP, improving AUC and calibration; early single-centre data suggest AFP-L3 ≥ 15% and DCP ≥ 7.5 ng/mL at LT enrich for early recurrence and may refine surveillance intensity and adjuvant-trial eligibility [115,116].

In practical terms, post-transplant surveillance should follow RETREAT-tiered imaging and AFP schedules, with mRETREAT markers such as AFP-L3 and DCP incorporated where available. Electronic alerts or structured reminders for high-risk strata (for example, RETREAT ≥ 3 or mRETREAT ≥ 4) can prompt shorter surveillance intervals, early PET or MRI assessment, and timely referral to recurrence-prevention or early-intervention trials.

### 8.3. Risk-Adapted Immunosuppression

Immunosuppression can be tailored to recurrence risk. The SiLVER randomised trial showed that sirolimus-based regimens did not improve long-term RFS beyond 5 years but yielded benefits in the first 3–5 years, particularly in lower-risk subsets—supporting early mTOR-inhibitor incorporation when recurrence risk is salient and renal/metabolic profiles permit [21]. Contemporary syntheses link mTOR inhibitors with lower early recurrence and better RFS/OS versus calcineurin inhibitor (CNI)-only strategies, acknowledging heterogeneity and the need for prospective confirmation [22]. Complementarily, early CNI minimisation (often IL-2RA-facilitated) has been associated with reduced recurrence in high-risk groups, though findings are mixed and should be balanced against rejection risk and centre experience [117]. In short, current consensus supports early mTOR-based regimens during the first 3–5 years in patients at high oncologic risk, with gradual CNI minimisation when feasible, and long-term maintenance tailored to graft function, metabolic safety, and comorbidity.

### 8.4. A Transplant-Aware Biomarker Workflow (Proposal)

The proposed transplant-aware workflow integrates morphological assessment, biological markers, and imaging at each phase of management. At the time of listing, candidates should first satisfy morphological criteria and undergo biological evaluation, including AFP trend, DCP or AFP-L3 level, PET avidity, and MVI-oriented radiomics. When findings are discordant or biologically unfavourable, locoregional therapy should be intensified, observation extended, or LDLT criteria that incorporate biological parameters, such as the Kyoto model, should be applied. Before exception approval, evidence of treatment response (mRECIST), stabilised tumour markers (for example, decreasing AFP), and restaging with CT or MRI are required, with PET or radiomics used to detect covert MVI. At transplantation, AFP should be recorded, MVI assessed in the explant, and viable tumour burden documented as inputs to the RETREAT or mRETREAT score. After transplantation, surveillance intensity should follow RETREAT (±mRETREAT) guidance; in high-risk strata, early mTOR-based immunosuppression and shorter imaging intervals may be appropriate, while CNI exposure should be minimised when clinically safe.

Concluding Remarks

Integrating liquid-biopsy-derived markers with established radiologic, serologic, and histopathologic variables provides a practical pathway toward biology-anchored transplant decision-making in HCC. Emerging workflows now incorporate ctDNA, methylation, and radiomics signatures alongside AFP, DCP, and PET findings to refine selection, risk stratification, and surveillance. Standardisation of assays and prospective validation of MRD-guided interventions will be essential to translate these tools from research frameworks into routine clinical algorithms.

## 9. Immunotherapy Around Transplant: Benefits and Risks

### 9.1. Pre-LT ICIs: Opportunity with a Time-Dependent Rejection Hazard

Why consider ICIs pre-LT? In selected candidates, ICIs can downstage/bridge to transplant. The central safety question is how long to wait between the last ICI dose and transplantation. An individual-patient-data meta-analysis of 91 HCC patients treated with ICIs before LT reported allograft rejection in 26.4% and showed that each additional week of washout lowered rejection risk (adjusted HR per week 0.92). A ~3-month washout brought the predicted rejection risk near baseline, with ~94 days corresponding to a ≤20% modelled risk [18].

Multicentre cohort data reinforce a 30-day threshold as a pragmatic minimum. In an 11-centre study (83 recipients), TLAT ≥ 30 days (time from last ICI to LT) independently protected against rejection (OR 0.096, 95% CI 0.026–0.357); TLAT < 30 days carried a markedly higher risk [118]. In a 2025 global cohort, the pre-LT analysis set (*n* = 204) showed higher graft rejection with washout ≤ 30 days vs. >30 days (33.3% vs. 15.2%), and with ≤1.5 ICI half-life counts vs. >1.5 (41.8% vs. 14.8%); multivariable models confirmed lower rejection risk with washout > 30 days (OR 0.36) and >1.5 half-life counts (OR 0.24) [119]. Expert commentaries converge on ≥30 days as a floor and often suggest half-life-based washouts (e.g., ≥ three half-lives) when feasible [120]. Emerging peri-operative series further flag washout < 30 days as an independent risk factor for peri-operative graft loss in ICI-related T-cell-mediated rejection [121].

In practical terms, for pre-LT management, when ICIs are used for bridging or downstaging, a structured approach is recommended. A minimum washout period of at least 30 days should be targeted, and where oncologically safe, this interval can be extended to approximately three months or 1.5–3 drug half-lives, depending on the specific ICI administered. Radiologic and serologic responses should be documented before submitting an exception request. Multidisciplinary discussion is strongly advised for any case in which the anticipated time to liver transplantation (TLAT) is shorter than 30–50 days.

### 9.2. Post-LT ICIs for Recurrence: Selective Use with Strict Monitoring

Pooled post-LT experience shows meaningful antitumour activity in a subset but substantial immunologic risk. A systematic review and pooled analysis of 52 patients reported acute rejection in 28.8% and graft-loss deaths in 13.4%. Disease control was achieved in approximately 44%, and responders lived longer than non-responders [122]. Early prospective work suggests risk may be stratifiable, in a single-centre pilot that pre-screened grafts for PD-L1 negativity, PD-1 blockade yielded 15% acute rejection (3/20) with exploratory survival signals—hypothesis-generating, not practice-changing [123]. Beyond programme-level experience, pharmacovigilance analyses indicate anti-PD-L1 may carry lower rejection risk than anti-PD-1 or CTLA-4, but these data are observational and confounded [124]. Time since LT may also matter; case series suggest earlier post-LT exposure carries a higher rejection risk than later use [125].

In practical terms, for post-LT management, ICIs should be used with caution. Monotherapy with PD-1 or PD-L1 blockade is preferred, while CTLA-4-containing regimens should generally be avoided. Risk–benefit discussions are essential, and laboratory or biopsy surveillance should be intensified during the first 6–8 weeks after treatment initiation. Graft PD-L1 immunohistochemistry may be considered when available, although PD-L1–guided selection has not yet been validated.

### 9.3. Representative Cases and Key Immunologic Issues

Severe early events can occur with short washout. A widely cited fatal case described massive hepatic necrosis after nivolumab stopped 8 days pre-LT [126]. More recently, a 2025 Frontiers case required rescue split LT for antibody-mediated rejection (AMR) after PD-1/PD-L1 exposure 16 days pre-LT, underscoring the potential for humoral as well as T-cell-mediated injury when washout is brief [127]. Ongoing correspondence in *Hepatology* highlights AMR after ICI as a recognised, if uncommon, entity [128].

AMR after LT is characterised by donor-specific antibodies (DSA), C4d deposition, and cholestatic graft injury, and may coexist with T-cell–mediated rejection. Management typically includes high-dose corticosteroids, plasmapheresis, intravenous immunoglobulin (IVIG), and anti-CD20 therapy, with proteasome or complement inhibitors reserved for refractory cases. Outcomes are variable, and the risk of retransplantation remains significant [127,129]. A recent *Frontiers* review emphasises early biopsy and routine DSA monitoring when enzymes rise after ICI [129]. Mechanistically, multiple series suggest graft PD-L1 expression may mark heightened rejection risk under PD-1 blockade, but prospective validation is limited [123,127].

### 9.4. Practice Points (For Multidisciplinary Tumour Boards)

In the pre-LT setting, when ICIs are used for downstaging or bridging, a washout period of more than 30 days should be targeted whenever possible. Ideally, a three-month or half-life-based interval tailored to the specific agent is recommended. Radiologic and serologic responses should be documented before requesting an exception, and intensified induction therapy (for example, IL-2RA) may be considered at transplantation if the washout period is short.

In the post-LT context, ICIs should be reserved for fit patients with limited therapeutic alternatives. PD-1 or PD-L1 monotherapy is preferred, and CTLA-4–containing regimens should generally be avoided. Graft PD-L1 testing may be considered when available, and close monitoring with early biopsy thresholds and routine DSA checks is advised.

When interpreting post-transplant ICI data, clinicians should recognise the influence of publication bias and heterogeneity in drug type, timing, and immunosuppression protocols. Participation in prospective registries or clinical trials is encouraged, and treatment decisions should always involve shared discussion of the potential risk of graft loss.

Concluding Remarks

The use of ICIs around LT offers clinical benefit in highly selected HCC patients but carries a measurable immunologic hazard.

Before transplantation, an adequate washout interval—ideally ≥ 30 days or approximately 3 months based on drug half-life—markedly reduces rejection risk.

After transplantation, ICIs should be restricted to fit recipients with limited options, favouring PD-1 or PD-L1 monotherapy with close laboratory and biopsy surveillance.

Antibody-mediated and T-cell-mediated rejection remain key concerns, emphasising the need for multidisciplinary oversight, standardised washout protocols, and prospective registry data to define safe practice boundaries.

## 10. Special Topics

### 10.1. ABO-Incompatible (ABOi) LDLT: Outcomes in the Rituximab Era

Rituximab-based desensitisation (±plasmapheresis) has transformed adult ABOi-LDLT from a high-risk to a clinically viable option. Contemporary meta-analyses and multicentre cohorts show patient/graft survival approaching ABO-compatible LDLT, while highlighting residual excess in antibody-mediated rejection (AMR), cytomegalovirus infection, and—most consistently—biliary complications (particularly diffuse intrahepatic biliary strictures, DIHBS). Mechanistically, isoagglutinin-mediated injury to the biliary epithelium and complement activation are implicated in this process. Protocol refinements (single-dose rituximab, tailored apheresis, local infusion strategies) have mitigated but not eliminated risk [130,131]. These data position biliary surveillance as a central component of post-transplant care in ABOi recipients.

ABOi-LDLT does not appear to increase HCC recurrence when rigorous oncologic selection and standard desensitisation are used. Propensity-matched and single-centre cohorts comparing ABOi versus ABO-compatible LDLT showed no significant differences in recurrence-free or overall survival, and ABO incompatibility was not an independent predictor of post-transplant recurrence after adjusting for tumour burden and biology [132,133,134]. Contemporary state-of-the-art reviews concur that rituximab-based desensitisation does not raise HCC recurrence risk [135].

In HCC candidates, rituximab-based protocols have not shown a clear signal for higher tumour recurrence versus ABO-compatible LDLT when selection and downstaging are rigorous. However, the biliary complication hazard remains a differentiator and should be incorporated into informed consent and follow-up planning.

### 10.2. Aetiology-Specific Biology (HBV/HCV/NASH): Molecular and Immune Heterogeneity with Therapeutic and Transplant Implications

The biology of HCC differs by aetiology. HBV-associated HCC often reflects viral–integration–driven oncogenesis (e.g., TERT activation) with distinctive immune contexts, whereas NASH-related HCC exhibits myeloid-skewed, T-cell-dysregulated microenvironments that can blunt immune surveillance. In preclinical and translational studies, NASH promotes the accumulation of dysfunctional PD-1^+^ CD8 T cells and can attenuate anti-PD-1 efficacy; HBV/HCV-related tumours display different interferon/inflammatory programmes and antigen-specific T-cell exhaustion patterns [136,137,138,139]. Clinically, multi-cohort analyses suggest non-viral HCC may have inferior outcomes on ICI versus viral etiologies, while trial-level subgroup results (e.g., IMbrave150) are exploratory and confounded by a non-stratified design—so aetiology alone should not dictate therapy but may inform expectations and trial stratification [138,140].

Among LT recipients with HCC, NASH cohorts are typically older with cardiometabolic multimorbidity, which influences peri- and post-transplant risk profiles (cardiovascular events, metabolic syndrome), even when oncological outcomes (OS/RFS) are broadly comparable to non-NASH etiologies across modern series [141,142]. These differences argue for a holistic, aetiology-aware approach. In NASH, prioritise cardiovascular optimisation before LT and during early post-LT follow-up. In HBV, maintain robust antiviral prophylaxis. In HCV-cured recipients, monitor for emerging metabolic risk factors.

Concluding Remarks

ABOi LDLT has become a clinically viable option in the rituximab era, achieving survival outcomes comparable to compatible LDLT when strict oncologic selection and standard desensitisation protocols are applied. Nonetheless, persistent biliary-complication risk warrants ongoing surveillance and protocol optimisation. Across aetiologies, viral- and metabolic-driven HCC exhibit distinct immune and molecular profiles that influence both systemic-therapy response and transplant outcomes. Future work should integrate aetiology-specific biology into selection and immunosuppression strategies to further personalise post-transplant care.

## 11. Proposed Practice-Oriented Integrated Algorithm

An integrated pathway for HCC management should combine stage, liver reserve, and tumour biology to align precision systemic options with transplant-based care (Figure 3). Initial triage should follow BCLC 2022, acknowledging heterogeneity within BCLC-B and the role of treatment stage migration in selected cases [3]. Liver functional reserve is quantified with ALBI rather than Child–Pugh to avoid subjective elements and to enable repeated, objective re-assessment as therapy proceeds [143,144].

At baseline—and at each decision point—biology should be layered on top of stage and liver function. Serologic profiling should include AFP (level and trajectory) with DCP (PIVKA-II) and AFP-L3, where available, given their orthogonal prognostic value and incorporation into living-donor selection frameworks such as Kyoto criteria [97,116,145]. Imaging phenotypes should be assessed using CT/MRI radiomics for pre-operative microvascular invasion prediction and 18F-FDG PET to flag aggressive biology that is not apparent on conventional morphology [13,106,108]. Genomic/TME features should inform regimen selection. WNT/CTNNB1 activation denotes an immune-excluded state associated with ICI resistance and should temper enthusiasm for checkpoint-heavy plans [9,10]. Where available, ctDNA should be incorporated as a molecular residual-disease read-out to anticipate relapse and enrich surveillance, with transplant-aware interpretation after LT [16,74,88].

Patients not immediately candidates for LT should enter a systemic-therapy branch. For advanced disease or “TACE-unsuitable” BCLC-B, first-line options include atezolizumab–bevacizumab, STRIDE (durvalumab plus tremelimumab), and nivolumab–ipilimumab following CheckMate-9DW. Selection should hinge on variceal/bleeding risk, autoimmune comorbidity, and tolerance for dual-IO toxicity [4,5,53]. In intermediate-stage disease where TACE remains appropriate, EMERALD-1 and LEAP-012 support adding IO (±VEGF/TKI) to TACE for PFS benefit, with centre-level adoption guided by OS maturity and grade ≥ 3 toxicity profiles [57,58]. Re-staging at programme-defined intervals—typically every 6–8 weeks during systemic therapy and after each TACE session—with ALBI recalculation and review of radiomics/PET and, where available, ctDNA dynamics should confirm benefit and safeguard liver reserve. When tumour biology remains favourable and hepatic function is preserved, stage migration toward the transplant pathway is warranted.

When downstaging to LT is pursued, selection should follow OPTN/UNOS standards: eligibility within UNOS-DS limits before locoregional therapy, fulfilment of T2 criteria by dynamic CT/MRI afterwards for a standardised exception, and reduction in AFP > 1000 ng/mL to <500 ng/mL sustained prior to exception; imaging terminology should conform to LI-RADS (including CEUS) [12,90,91]. Response quality should influence listing decisions: incorporating mRECIST into Metroticket 2.0 improves prediction of HCC-related death, and poor or unstable response should trigger stricter gates or extended observation [99]. Appropriately selected candidates can be successfully downstaged in >80% of cases, with post-LT outcomes approaching those of within-Milan patients when downstaging succeeds [92,146]. In instances of discordance between morphology and biology—such as within-Milan tumours with high PET avidity and rising AFP/DCP—priority should be given to biology (e.g., intensified bridging or longer observation).

After transplantation, risk-adapted surveillance and immunosuppression should be implemented. The RETREAT score—comprising AFP at LT, MVI, and viable tumour burden—should be used to stratify 5-year recurrence risk (from less than 3% at score 0 to more than 75% at score ≥ 5) and to determine imaging and AFP surveillance cadence. External validations and prospective multicentre data support programme-level adoption [112,114]. ctDNA may serve as an adjunct in higher-risk strata, with positives corroborated given transplant-specific analytical caveats (e.g., donor-derived cfDNA) [16,88]. For immunosuppression, early mTOR-inhibitor incorporation during the first 3–5 years may be considered for patients with salient oncological risk, and CNI exposure should be minimised where clinically safe; durable long-term RFS advantage has not been demonstrated, so choices should be individualised to graft function and comorbidity [21,22,117].

Overall, the proposed pathway emphasises cyclical assessment: stage and ALBI anchor feasibility, biology refines direction and observed response re-opens options. Patients move between three coordinated lanes—precision systemic therapy ± TACE, downstaging to LT, and post-LT risk-adapted care—with each lane informed by the same set of inputs and bounded by policy and safety constraints.

## 12. Conclusions and Key Messages

Precision medicine in HCC is moving from drug choice to treatment architecture. Molecular and immune markers—serology (AFP, DCP, AFP-L3), imaging phenotypes (radiomics for microvascular invasion, FDG-PET avidity), genomic/TME features (e.g., CTNNB1/WNT activation), and liquid biopsy (ctDNA/MRD)—are increasingly actionable when embedded directly into transplant decisions. Coupled with BCLC 2022s endorsement of stage migration, UNOS/OPTN policy harmonisation with LI-RADS, and stronger first-line systemic options, these tools enable a transplant-aware precision pathway that prioritises biology as much as morphology.

Key messages

Biology plays a decisive role at the time of listing. AFP dynamics, DCP/AFP-L3, PET avidity, and MVI-oriented radiomics help identify high-risk tumour biology within or beyond morphologic criteria and should guide the intensity of down-staging, observation periods, and exception requests. Stage migration should be deliberate: restaging at programmed intervals with ALBI recalculation and biomarker or imaging review prevents futile therapy and ensures timely transition between systemic and transplant pathways.

In systemic therapy, multiple first-line immunotherapy combinations—including atezolizumab plus bevacizumab, STRIDE, and nivolumab plus ipilimumab—offer validated efficacy; regimen choice depends on bleeding risk, autoimmune comorbidity, and tumour biology. IO/TACE combinations show progression-free-survival benefit, but adoption should proceed cautiously pending mature overall-survival data. Adjuvant or neoadjuvant systemic therapy is not yet standard; surveillance remains the preferred approach, and clinical-trial enrolment is encouraged.

Successful down-staging under UNOS-DS criteria can yield post-transplant outcomes comparable to within-Milan candidates when biological features are favourable. Post-transplant care should be risk-adapted using the RETREAT (±mRETREAT) framework with ctDNA adjuncts in high-risk strata. Immunosuppression optimisation—early mTOR-inhibitor incorporation and cautious CNI minimisation—may reduce early recurrence, though long-term benefit remains uncertain. ICIs around transplantation require careful timing and multidisciplinary oversight, while ABOi-LDLT has become oncologically acceptable in the rituximab era despite residual biliary-complication risk.

What Should Change Next?

Future directions span trials, policy, and clinical practice. In clinical research, priority should be given to MRD-guided perioperative strategies, biology-stratified systemic regimens—including WNT/CTNNB1-informed trial designs—and prospective algorithms that integrate radiomics, PET, and ctDNA into listing, exception, and surveillance pathways. In policy, continued alignment of allocation rules with tumour biology is essential, incorporating AFP- and DCP-anchored thresholds, response-weighted exceptions, and harmonised LI-RADS imaging language across jurisdictions. In everyday practice, routine multidisciplinary case reviews combining stage, ALBI, and biomarker panels, along with centre-specific restaging schedules and structured post-transplant pathways linked to RETREAT tiers (with optional ctDNA monitoring), will help embed a truly biology-driven workflow into transplant care.

A coherent precision-and-transplant framework is now feasible: stage and ALBI anchor clinical feasibility, biology refines direction, and response reopens options. Embedding this iterative cycle into everyday workflows will help close the gap between clinical trials, policy, and bedside care.

## Figures and Tables

**Figure 1 cancers-17-03671-f001:**
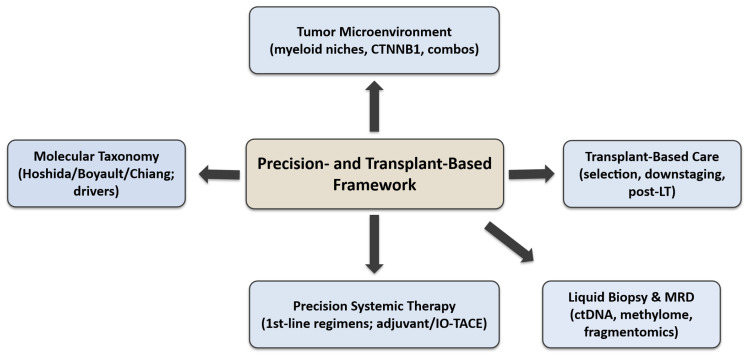
Overview of key molecular, immunological, and transplant-based aspects discussed in this review.

**Figure 2 cancers-17-03671-f002:**
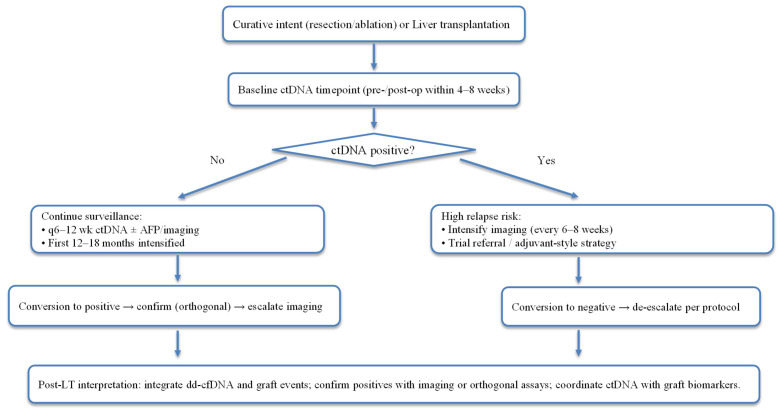
ctDNA/MRD-aware surveillance around curative therapy and liver transplantation.

**Figure 3 cancers-17-03671-f003:**
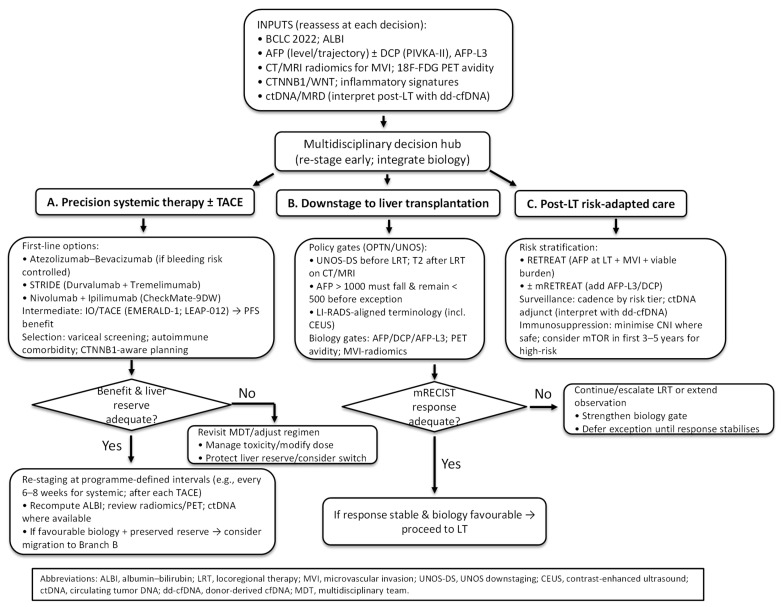
Integrated decision tree for precision and transplant-based care in HCC.

**Table 1 cancers-17-03671-t001:** Molecular taxonomies of HCC and clinical correlates (summary).

Framework	Biology (Signals)	Key Drivers	Clinical Correlates	Refs.
Hoshida S1–S3	S1: WNT/TGF-β; S2: proliferation (MYC/AKT); S3: hepatocytic differentiation	CTNNB1 (S3 low), MYC/AKT (S2)	S2 larger tumours, higher AFP; S3 lower AFP, better differentiation; S1 early recurrence	[23]
Boyault G1–G6	HBV/AKT-PI3K (G1/2); cell-cycle/TP53 (G3); TCF1 (G4); β-catenin (G5/6)	TP53, PIK3CA, TCF1, CTNNB1	G6 E-cadherin loss, satellites; genotype–phenotype mapping	[24]
Chiang (5 classes)	CTNNB1; proliferation; IFN-related; chr7 polysomy; unannotated	CTNNB1, IGF/AKT-mTOR	Expression–copy-number integration	[25]
Integrative clinical view	Non-proliferation (often CTNNB1) vs. Proliferation (TERT-p, TP53, FGF19/FGFR4, CIN)	TERT-p, TP53, FGF19 amp/FGFR4	Bedside-usable dichotomy guiding therapy hypotheses	[27,28,29,30,31]

Notes: Proteogenomics shows transcript–protein discordance and pathway-level rewiring, supporting multi-omics profiling where feasible [32,33].

**Table 2 cancers-17-03671-t002:** Key first-line and second-line systemic-therapy trials for unresectable/advanced HCC.

Regimen/Trial	Comparator	Median OS (Months)	HR (OS)	Median PFS (Months)	Key Toxicities	References
Atezolizumab + Bevacizumab (IMbrave150)	Sorafenib	19.2 vs. 13.4	0.66	6.9 vs. 4.3	Hypertension, bleeding risk	[4,7]
Tremelimumab + Durvalumab (STRIDE/HIMALAYA)	Sorafenib	16.4 vs. 13.8	≈0.78	3.8 vs. 4.1	Immune-related AEs (CTLA-4 priming)	[5,52]
Nivolumab + Ipilimumab (CheckMate 9DW)	Lenvatinib/Sorafenib	23.7 vs. 20.6	0.79	(OS primary)	Immune-related AEs	[53]
Regorafenib (RESORCE)	Placebo (after Sorafenib)	10.6 vs. 7.8	0.63	3.1 vs. 1.5	Hand–foot reaction, fatigue	[54]
Cabozantinib (CELESTIAL)	Placebo (after Sorafenib)	10.2 vs. 8.0	0.76	5.2 vs. 1.9	Diarrhoea, hypertension, fatigue	[55]
Ramucirumab (REACH-2)	Placebo (AFP ≥ 400 ng/mL)	8.5 vs. 7.3	~0.71	2.8 vs. 1.6	Hypertension, proteinuria	[56]
Durvalumab + Bevacizumab + TACE (EMERALD-1)	TACE + Placebo	(OS immature)	—	15.0 vs. 8.2	Hypertension, elevated ALT	[57]
Lenvatinib + Pembrolizumab + TACE (LEAP-012)	TACE + Placebo	(OS immature)	—	14.6 vs. 10.0	Hypertension, immune-related AEs	[58]

Abbreviations: OS, overall survival; PFS, progression-free survival; HR, hazard ratio; AE, adverse event; TACE, transarterial chemoembolisation.

**Table 3 cancers-17-03671-t003:** Ongoing and recently reported (neo)adjuvant trials.

Trial (Phase)	Setting/Regimen	Key Population and Timing	Primary End Point/Status (as of Sept 2025)	Notes
IMbrave050 (III; NCT04102098)	Adjuvant atezolizumab + bevacizumab vs. active surveillance	High-risk after R0 resection/ablation; start ≤ 12 weeks	Updated: RFS not sustained (HR 0.90); OS immature	Not recommended for universal use per AASLD 2025 [8,66].
EMERALD-2 (III; NCT03847428)	Adjuvant durvalumab ± bevacizumab vs. placebo	High-risk after resection/ablation	Ongoing; results pending	Trial design available; no peer-reviewed outcomes yet.
CheckMate-9DX (III; NCT03383458)	Adjuvant nivolumab vs. placebo	High-risk after resection/ablation	Ongoing; results pending	Global, double-blind study.
KEYNOTE-937 (III; NCT03867084)	Adjuvant pembrolizumab vs. placebo	CR after resection/ablation	Ongoing; results pending	Tissue-agnostic approvals do not extrapolate to adjuvant HCC.
SHR-1210-III-325 (III; NCT04639180)	Adjuvant camrelizumab + rivoceranib (apatinib) vs. placebo	High-risk after resection/ablation	Ongoing (China); status “unknown” on registries	Non-global sponsor; watch for regional readouts.
PRIME-HCC/peri-op nivolumab ± ipilimumab (Ib/II)	Neoadjuvant/peri-op	Resectable HCC	Early-phase feasibility signals	Hypothesis-generating only [69].
Cabozantinib + nivolumab (Ib; NCT03299946)	Neoadjuvant combo	Borderline-resectable	Early-phase conversion to resectability	Supports further testing [70].

Sources for the table details: IMbrave050 initial and updated reports; AASLD 2025 update; ClinicalTrials.gov and sponsor protocols for EMERALD-2, CheckMate-9DX, KEYNOTE-937; registry listings for camrelizumab + rivoceranib; PRIME-HCC and cabozantinib + nivolumab early-phase publications.

**Table 4 cancers-17-03671-t004:** OPTN/UNOS HCC policies relevant to downstaging and exception (snapshot).

Domain	Policy Elements (Abridged)	Practical Notes	Refs.
Standardised exception (T2)	1 lesion 2–5 cm or 2/3 lesions 1–3 cm; AFP ≤ 1000 ng/mL	Dynamic CT/MRI using LI-RADS; CEUS accepted	[89,90]
UNOS-DS (pre-LRT)	1 lesion 5–8 cm; or 2/3 lesions (≤5 cm, sum ≤ 8 cm, ≥1 > 3 cm); or 4/5 lesions (< 3 cm, sum ≤ 8 cm). After LRT, must meet T2 for exception		[89]
AFP rule	If AFP > 1000, treat; exception requires AFP < 500 and sustained	Safeguard against aggressive biology	[12,89]
Allocation lexicon	LI-RADS alignment (since 13 July 2023)	Harmonises radiology reporting across centres	[90]

Notes: With UNOS-DS, > 80% can be downstaged successfully; post-LT outcomes approximate within-Milan when response is adequate. [91,92,93].

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
