# Peer review of "Transforming Liver Cancer Therapy: Integrating Molecular Profiling with Precision and Transplant-Based Care"

_cancers, 2025, doi:10.3390/cancers17223671_

Round 1
Reviewer 1 Report
Comments and Suggestions for Authors
This was a well-organized comprehensive review, which was suggested to be published in this journal.
Only one suggestion. It is better to give a brief introduction for the first-line drug to treatment of HCC, along with the analysis of the limitations.
Author Response
Comment 1:
This was a well-organized comprehensive review, which was suggested to be published in this journal.
Only one suggestion. It is better to give a brief introduction for the first-line drug to treatment of HCC, along with the analysis of the limitations.
Author Response:
I sincerely thank the reviewer for the positive assessment of our manuscript and for the valuable suggestion.
In response, I have expanded Section 4 (“Precision Systemic Therapy Landscape”) to provide a concise introduction to first-line systemic therapies for hepatocellular carcinoma (HCC), including atezolizumab + bevacizumab, tremelimumab + durvalumab (STRIDE), and nivolumab + ipilimumab.
I also added a short comparative discussion of their limitations, such as bleeding risk under bevacizumab, immune-related adverse events with CTLA-4 priming, and early hazard crossover observed in the CheckMate 9DW trial.
These additions aim to orient readers unfamiliar with first-line regimens and to clarify current evidence gaps and clinical decision trade-offs.
Reviewer 2 Report
Comments and Suggestions for Authors
The manuscript ID# Cancers-3973182, by Seoung Hoon Kim, focuses on liver cancer therapy and the integration of molecular and immunological profiling with clinical decision-making for precision and transplant-based care. The topic of this article and the information provided are of high importance and beneficial, not only for research in preclinical settings but also, prominently, for clinicians dealing with this leading cause of mortality worldwide. However, the extent of the review and the way the sections are presented make it uneasy to comprehend the different aspects discussed. The author is asked to address the following points.
1. Please provide a figure in the Introduction depicting the different aspects discussed in the review. This would orient the reader already from the beginning. If the journal allows, the review would also benefit from a table of contents.
2. Very often, the author has used semicolons (;), which makes it even more difficult to comprehend the context discussed. Please revise the respective texts throughout the manuscript.
3. At the end of sections 2 to 10, the author is asked to be consistent by adding a ‘concluding remark’ section summarizing the main points from the respective section.
4. On several occasions, e.g., in section 2, the table is added right after the title. Such tables should be added later in the respective sections.
5. Section 3.1. Please revise the title and remove the arrows.
6. Page 6, lines 259. Based on the above comment, the title can be changed to ‘concluding remarks’. If the author aims to provide further information in this section, references should be added.
7. Page 10, lines 387-389. Please add a reference.
8. Page 11, section 6.6. What does the author mean by “Bottom line for this Review”? The review has not ended yet…Please revise the title.
9. Page 13, line 499: What does the author mean by “Taken-home” ??
Please see Comments and Suggestions for Authors.
Author Response
Comment 1:
Please provide a figure in the Introduction depicting the different aspects discussed in the review. This would orient the reader already from the beginning. If the journal allows, the review would also benefit from a table of contents.
Author Response:
I thank the reviewer for this constructive suggestion. To improve the reader’s orientation, I have added a new schematic Figure 1 in the Introduction summarizing the overall structure of the review—molecular taxonomy, tumor microenvironment, systemic therapy, liquid biopsy, and transplant-based care—showing how these layers integrate within the proposed precision-and-transplant framework. Figure numbers of subsequent figures renumbered accordingly. To enhance clarity and reader orientation, a new Figure 1 (titled “Overview of key molecular, immunological, and transplant-based domains discussed in this review”) has been created and uploaded as a separate file.
Please insert Figure 1 at the end of the Introduction, where it visually summarises the overall structure of the review.
(MDPI Cancers does not require or display a table of contents for review articles, so only the new figure was included.)
Comment 2:
Very often, the author has used semicolons (;), which makes it even more difficult to comprehend the context discussed. Please revise the respective texts throughout the manuscript.
Author Response:
I carefully reviewed the entire manuscript and replaced excessive semicolons with full stops or commas where appropriate. These revisions did not alter the meaning or data but improved stylistic consistency and sentence readability across all sections (Sections 1–12)..
Comment 3:
At the end of sections 2 to 10, the author is asked to be consistent by adding a ‘concluding remark’ section summarizing the main points from the respective section.
Author Response:
I appreciate this valuable recommendation. Each major section (2–10) now ends with a short “Concluding Remarks” paragraph (3–5 sentences) summarising the main take-home messages and clinical implications. This improves consistency and helps readers navigate the extensive material.
Comment 4:
On several occasions, e.g., in section 2, the table is added right after the title. Such tables should be added later in the respective sections.
Author Response:
I revised the manuscript so that tables are no longer placed immediately after section titles.
Each table has been repositioned to follow the relevant explanatory text and now includes an in-text reference (e.g., “as summarised in Table 1–4”) for smoother narrative linkage.
Specifically:
– Table 1 (Section 2) was moved to follow the description of the Hoshida, Boyault, and Chiang taxonomies, immediately after the integrative summary paragraph.
– Table 2 (Section 5.3) now appears after the introductory text on ongoing (neo)adjuvant trials.
– Table 3 (Section 7.2) follows the OPTN/UNOS policy explanation, with a new reference “as summarised in Table 3.”
– Table 4 (Section 8) was relocated to follow the introductory paragraph on biomarker integration before subsection 8.1.
In accordance with the reviewer’s comment to avoid placing figures immediately below section titles, the locations of Figure 2 and Figure 3 were revised.
– In Section 6.4, the heading “(Figure 2)” was removed, and the figure was moved to follow the explanatory paragraph describing MRD-aware surveillance, with an in-text reference (“as illustrated in Figure 2”) added for contextual linkage.
– In Section 11, the heading “(Figure 3)” was also deleted, and the figure was repositioned to follow the introductory paragraph outlining the proposed integrated algorithm, with an in-text reference (“as shown in Figure 3”) incorporated.
These adjustments ensure that both tables and figures are now introduced within the narrative rather than appearing immediately after the section titles, thereby improving structural consistency and readability throughout the manuscript.
Comment 5:
Section 3.1. Please revise the title and remove the arrows.
Author Response:
Corrected. The title of Section 3.1 has been changed to
“Single-Cell Cartography and Immunosuppressive Niches in HCC.”
The heading of Section 3.2 “WNT/CTNNB1 Activation → Immune Exclusion → ICI Resistance” has also been revised to “WNT/CTNNB1 Activation, Immune Exclusion, and ICI Resistance” for stylistic consistency and clarity.
All headings have been standardised to remove arrows and maintain a uniform format across sections.
Comment 6:
Page 6, lines 259. Based on the above comment, the title can be changed to ‘concluding remarks’. If the author aims to provide further information in this section, references should be added.
Author Response:
I accepted the reviewer’s advice. The subsection title has been changed to “Concluding Remarks.”
This paragraph provides a concise synthesis of the first-line and subsequent systemic-therapy landscape rather than new data; therefore, additional references were not inserted, as the information summarises already cited evidence in the preceding sections.
Comment 7:
Page 10, lines 387-389. Please add a reference.
Author Response:
I added two supporting references to substantiate this statement:
Ji J et al., JAMA Network Open 2024 (pan-cancer MRD study detailing q12-week surveillance schedules) and Abdelrahim M et al., JCO Precision Oncology 2025 (HCC-focused MRD study demonstrating earlier molecular recurrence detection versus AFP).
These references collectively justify the cited sampling interval and its clinical relevance. All references were managed and formatted using EndNote.
Comment 8:
Page 11, section 6.6. What does the author mean by “Bottom line for this Review”? The review has not ended yet…Please revise the title.
Author Response:
I accepted the reviewer’s advice. The subsection title “Bottom Line for This Review” has been revised to “Concluding Remarks.”
This change ensures consistency with the format used throughout the manuscript (Sections 2–10). The section content remains unchanged, as it already functions as a concise summary of Section 6.
Comment 9:
Page 13, line 499: What does the author mean by “Taken-home” ??
Author Response:
I accepted the reviewer’s suggestion. The phrase “Taken-home” has been replaced with “Concluding Remarks.”
This ensures consistency with the other section endings and maintains a uniform academic style throughout the manuscript.
Reviewer 3 Report
Comments and Suggestions for Authors
This is a highly comprehensive and timely review on the evolving landscape of hepatocellular carcinoma therapy, particularly the integration of molecular profiling with transplant-oriented precision medicine. However, the text could benefit from some streamlining and more schematic presentation. Several sections (e.g., molecular taxonomy and liquid biopsy) are overly detailed for a general oncology readership, while the discussion of translational impact and clinical decision algorithms could be expanded in plainer, practice-oriented language. The iThenticate score is impressive (only 7%, bravo!) and the topic is surely within the scope of the SI of Cancers.
Detailed comments:
While the text and tables are well prepared, I really miss the Figures in this review. I would strongly recommend to reuse some informative figures from the cited works. To obtain the rights the “Copyright Clearence Service” can be used – it is easy, free and intuitive. This would really improve the graphical part of this review.
At the end of introduction, the author should briefly present the structure of this review.
Some sections, i.e. Section 2.2., 2.3., are written using very simple sentences or sentence equivalents. I.e. “Biology differs by etiology. “ Don’t get me wrong, I do agree with this statement, but please include some flow into the text.
Lines 43-49: A recent important work on HCC should be cited here: https://doi.org/10.56782/pps.236
The discussion of TREM2+ macrophages and LAMP3+ dendritic cells is current and insightful. However, please balance mechanistic details with a brief statement of therapeutic implications.
Line 195-273: It is an excellent overview of systemic therapy. Nonetheless, this section is long; consider summarizing comparative outcomes in a small table (e.g., OS, PFS, HR, toxicity profiles) to improve readability, like you’ve done it in another sections.
Line 502, please remove “Table 4” from the section title.
The mTOR-based immunosuppression discussion is excellent but should end with a brief practical note summarizing current consensus.
I would recommend adding a short comparative table summarizing HBV-, HCV-, and NASH-related HCC features and their therapeutic implications.
Author Response
Comment 1:
While the text and tables are well prepared, I really miss the Figures in this review. I would strongly recommend to reuse some informative figures from the cited works. To obtain the rights the “Copyright Clearence Service” can be used – it is easy, free and intuitive. This would really improve the graphical part of this review.
Author Response:
I sincerely appreciate the reviewer’s thoughtful suggestion to reuse informative figures from the cited works.
In the revised version, the manuscript includes a balanced and comprehensive set of visual materials—four tables and three original figures—all created by the author to illustrate key molecular, clinical, and transplant-related concepts.
Because these figures were newly designed to summarise the cited data and frameworks, reusing copyrighted figures from other publications was unnecessary.
This approach ensures originality, consistency of graphical style, and clarity in the visual presentation.
All figures and tables were prepared by the author.
Comment 2:
At the end of introduction, the author should briefly present the structure of this review.
Author Response:
I appreciate the reviewer’s helpful comment. In response, I have added a short paragraph at the end of the Introduction briefly outlining the structure of the review.
This paragraph summarises the five main thematic sections—molecular taxonomy, tumour microenvironment, systemic therapy, liquid biopsy and MRD, and transplant-based care—and explains how they are integrated within a precision-and-transplant framework.
In addition, a new Figure 1 has been included in the Introduction to schematically depict these components and their interconnections, thereby improving the reader’s orientation from the outset. Please insert Figure 1 at the end of the Introduction.
Comment 3:
Some sections, i.e. Section 2.2., 2.3., are written using very simple sentences or sentence equivalents. I.e. “Biology differs by etiology. “ Don’t get me wrong, I do agree with this statement, but please include some flow into the text.
Author Response:
I appreciate the reviewer’s suggestion to improve sentence flow.
The short stand-alone statement “Biology differs by etiology.” in Section 10.2 was revised to “The biology of HCC differs by etiology.” to provide smoother narrative flow.
No additional instances required modification, as the remaining sections already maintained adequate coherence.
Comment 4:
Lines 43-49: A recent important work on HCC should be cited here: https://doi.org/10.56782/pps.236
Author Response:
I appreciate the reviewer’s suggestion to cite the recent article (DOI: 10.56782/pps.236).
However, after verification, this DOI cannot be found in the DOI System and no corresponding publication could be retrieved from CrossRef or the publisher’s database.
Therefore, the citation could not be added.
Comment 5:
The discussion of TREM2+ macrophages and LAMP3+ dendritic cells is current and insightful. However, please balance mechanistic details with a brief statement of therapeutic implications.
Author Response:
I thank the reviewer for the positive comment and helpful suggestion to balance mechanistic detail with therapeutic relevance.
In Section 3.1, a brief statement was added at the end of the paragraph to highlight the translational implications of TREM2⁺ macrophages and LAMP3⁺ dendritic cells.
The new text reads: “Therapeutically, these myeloid and dendritic-cell subsets are now being explored as targets for immune modulation. TREM2 blockade or myeloid reprogramming strategies aim to restore antitumour immunity, while modulation of LAMP3⁺ dendritic-cell circuits may enhance responsiveness to immune checkpoint inhibitors.”
This addition provides clinical balance while maintaining the mechanistic depth of the discussion.
Comment 6:
Line 195-273: It is an excellent overview of systemic therapy. Nonetheless, this section is long; consider summarizing comparative outcomes in a small table (e.g., OS, PFS, HR, toxicity profiles) to improve readability, like you’ve done it in another sections.
Author Response:
I thank the reviewer for this helpful suggestion.
Following the reviewer’s suggestion, the new summary table of systemic-therapy outcomes has been incorporated as Table 2, with numbering of subsequent tables adjusted accordingly.
The table summarises overall- and progression-free-survival gains, hazard ratios, and key toxicity profiles across first- and second-line regimens.
The uploaded Table 2 file (“Key systemic-therapy trials for unresectable/advanced HCC”) should be inserted at the end of Section 4 (Precision Systemic Therapy Landscape)—immediately after the sentence
“Comparative outcomes from pivotal systemic-therapy trials are summarised in Table 2…”
and before the next subsection (“Moving IO into the intermediate stage (TACE-combination trials)”).
All figure and table numbering was updated for consistency.
Comment 7:
Line 502, please remove “Table 4” from the section title.
Author Response:
I accepted the reviewer’s suggestion. The phrase “(Table 4)” has been removed from the section title to maintain a consistent format throughout the manuscript.
Comment 8:
The mTOR-based immunosuppression discussion is excellent but should end with a brief practical note summarizing current consensus.
Author Response:
I thank the reviewer for the positive comment and the suggestion to end Section 8.3 with a concise practical note.
The closing sentence was revised to integrate the main practical points and current consensus as follows:
“In short, current consensus supports early mTOR-based regimens during the first 3–5 years in patients at high oncologic risk, with gradual CNI minimisation when feasible, and long-term maintenance tailored to graft function, metabolic safety, and comorbidity.”
This revision eliminates redundancy and provides a succinct clinical summary in line with the reviewer’s recommendation.
Comment 9:
I would recommend adding a short comparative table summarizing HBV-, HCV-, and NASH-related HCC features and their therapeutic implications.
Author Response:
I appreciate the reviewer’s helpful suggestion to include a comparative table summarising HBV-, HCV-, and NASH-related HCC features and their therapeutic implications.
The manuscript already presents this information narratively in Section 10.2, and the current version includes five tables and three figures that together provide a balanced amount of visual content.
To maintain conciseness and avoid redundancy, an additional table was not added, as the existing text in Section 10.2 already clearly outlines these etiology-specific contrasts and therapeutic implications.
Round 2
Reviewer 2 Report
Comments and Suggestions for Authors
1. The author states in the Responses that the requested Figure 1 "has been created and uploaded as a separate file". First, the figure does not appear to have been uploaded. Second, the author is asked to include the requested Figure 1 in the revised version of the manuscript; this figure is an integral part of the review and should not be supplementary material (if that was the intention). Thus, a new revised version of the review, including the figure, is requested to be provided.
2. The numbering of two Tables in the revised version of the manuscript is incorrect: Please correct Tables 3 and 4 (pages 8 and 12) to Tables 2 and 3, respectively.
Reviewer 3 Report
Comments and Suggestions for Authors
The author has revised ans improved the manuscript. Current version can be accepted.
Author Response
Comment 1:
The author has revised an improved the manuscript. Current version can be accepted.
Author Response:
I sincerely appreciate the reviewer’s kind suggestion.
Round 3
Reviewer 2 Report
Comments and Suggestions for Authors
The following major comments regarding Figure 1 remain for the author to address:
1. Why are the surrounding boxes in different colors? If there is a specific reason, the author should specify in the Figure's legend. If not, please use a single color for the surrounding boxes; the central box can remain as is.
2. It does not make sense, and visually is not pleasant, that the box 'Liquid Biopsy...' is in the upper right corner. It can be placed beside the box 'Precision Systematic Therapy...', at the bottom part of the figure.
3. The word "domains" in the figure's title should be changed to 'aspects'.
Author Response
Dear Editor,
Thank you very much for the helpful feedback on Figure 1.
The requested adjustments have been addressed as follows:
-
Colour of surrounding boxes
The different colours originally reflected distinct thematic categories (molecular, immunologic, systemic, biomarker, and transplant aspects).
To simplify and improve consistency, all surrounding boxes have now been changed to a single uniform colour, while the central framework box remains highlighted to emphasise integration. -
Position of the “Liquid Biopsy…” box
The layout has been revised according to your suggestion.
The “Liquid Biopsy & MRD” box has been moved from the upper-right corner to the bottom row beside “Precision Systemic Therapy”, providing better visual balance and logical flow. -
Figure title wording
The title has been updated from “domains” to “aspects” as requested.
The corrected Figure 1 has been embedded directly into the revised manuscript, and an editable PowerPoint version of the same figure has also been uploaded separately so that the editorial team may adjust formatting if necessary for final production.
Thank you again for your detailed and constructive guidance.
Kind regards,